# Ultrasonic barrier-through imaging by Fabry-Perot resonance-tailoring panel

Chung Il Park [1,2], Seungah Choe[1,2], Woorim Lee[1,2], Wonjae Choi[3,4], Miso Kim [5,6], Hong Min Seung [3,4] ✉ & Yoon Young Kim [1,2] ✉

Imaging technologies that provide detailed information on intricate shapes and states of an object play critical roles in nanoscale dynamics, bio-organ and cell studies, medical diagnostics, and underwater detection. However, ultrasonic imaging of an object hidden by a nearly impenetrable metal barrier remains intractable. Here, we present the experimental results of ultrasonic imaging of an object in water behind a metal barrier of a high impedance mismatch. In comparison to direct ultrasonic images, our method yields sufficient object information on the shapes and locations with minimal errors. While our imaging principle is based on the Fabry-Perot (FP) resonance, our strategy for reducing attenuation in our experiments focuses on customising the resonance at any desired frequency. To tailor the resonance frequency, we placed an elaborately engineered panel of a specific material and thickness, called the FP resonance-tailoring panel (RTP), and installed the panel in front of a barrier at a controlled distance. Since our RTP-based imaging technique is readily compatible with conventional ultrasound devices, it can realise underwater barrier-through imaging and communication and enhance skull-through ultrasonic brain imaging.

Imaging providing hidden information on intricate shapes[1–3] and states[4–6] of a target object is indispensable in nanoscale dynamics[1], bio-organ[2] and cell studies[3], medical diagnostics[4,5], and underwater detection[7]. Advanced techniques[8–10] can substantially improve the quality of the acquired amplitude/phase information from an object behind an aberrating medium. Various imaging techniques have been used to measure waves interacting with target objects[4,5,8–10]. These waves carry object information in various forms from which the image of an object can be constructed. Highly informative images cannot be captured unless the measured wave fields are sufficiently intact from disturbances. However, various interfering sources, such as dissimilar media[8–13] placed in front of the target object, can severely distort the spatial amplitude and/or phase profiles of the measured wave field. A

considerable amount of effort has been spent on overcoming these difficulties, including those regarding advanced wave signal processing[8–10,14]. For instance, additional apparatuses or synthetic apertures[14,15] were used to compensate for aberrations by barriers[10,16,17] in the electromagnetic wave regime. Similar efforts have been made using acoustic waves in medical[18] and underwater imaging[19,20]. However, these approaches typically use additional bulky devices or complicated sensing processes that are not directly applicable to the ultrasonic imaging of objects behind highly impedance-contrasted barriers. Recently proposed metamaterials that enable elaborate wave controls[11,12,21–23] may be a promising solution. If a complementary metamaterial with effective negative material properties is placed in front of a barrier, it can theoretically nullify the distortion by a barrier,

[1]Department of Mechanical Engineering, Seoul National University, 1 Gwanak-ro, Gwanak-gu, Seoul 08826, Republic of Korea. [2]Institute of Advanced Machines and Design, Seoul National University, 1 Gwanak-ro, Gwanak-gu, Seoul 08826, Republic of Korea. [3]Intelligent Wave Engineering Team, Korea Research Institute of Standards and Science (KRISS), 267 Gajeong-ro, Yuseong-gu, Daejeon 34113, Republic of Korea. [4]Department of Precision Measurement, University of Science and Technology (UST), 217 Gajeong-ro, Yuseong-gu, Daejeon 34113, Republic of Korea. [5]School of Advanced Materials Science and Engineering, Sungkyunkwan University (SKKU), 2066 Seobu-ro, Jangan-gu, Suwon 16419, Republic of Korea. [6]SKKU Institute of Energy Science and Technology (SIEST), Sungkyunkwan University (SKKU), 2066 Seobu-ro, Jangan-gu, Suwon 16419, Republic of Korea. ✉e-mail: shm@kriss.re.kr; yykim@snu.ac.kr

allowing full wave transmission[11,12]. However, the required negative effective material properties are attainable using local dipole and monopole resonances, which are highly susceptible to fabrication errors. This challenges the application of metamaterials in barrier-through imaging. Other metamaterial-based approaches for extra-ordinary transmission[21,22] have similar problems. Therefore, a highly transmissive yet realisable imaging method must be devised to image the detailed geometry of an object behind a nearly impene-trable barrier. The problem considered here is the ultrasonic ima-ging of a two-dimensional object in water hidden behind a highly impedance-contrasted metal barrier, such as a ship hull wall of a sunken vessel.

In this study, we present a robust method aimed at demonstrating barrier-through ultrasonic imaging of the objects hidden by a nearly impenetrable barrier of a high impedance mismatch. Our approach leverages the Fabry-Perot (FP) resonance phenomena to achieve full transmission through a barrier. Diverging from conventional FP reso-nance, we introduce an additional element—the FP resonance-tailoring panel (RTP)—positioned at a controlled distance, which achieves full transmission through the barrier by adjusting the multiple scattering between the panel and the barrier. Through careful design of the impedance and thickness of the RTP, we demonstrate ultrasonic ima-ging through a nearly impenetrable metal barrier of a high impedance mismatch. The quality of the ultrasonic images obtained with the RTP is nearly equivalent to those obtained without any barriers, under-scoring the validity of our approach. Additionally, we demonstrate the simulation of ultrasonic beam transmission through a barrier using the RTP, further validating our proposition and presenting a possible potential application.

## Results

### Transmission enhancement through a barrier by the RTP

Figure 1a depicts a schematic illustration of the ultrasonic inspection within a submerged vessel, where the zoomed view conceptually compares ultrasound images through the metal hull wall with and without using the FP RTP we propose in this study; the metal wall has a highly contrasted impedance compared to water. Although the working principle of the RTP and the actual experimental results are given in Figs. 2 and 3, Fig. 1a suggests that without the RTP, the reflected signals from an object inside the vessel hull carry virtually no information on the target object. Figure 1b shows the attenuation of the ultrasound amplitude in the water[24,25] as a function of the travel distance ($L$) with and without a barrier. The signal received by an ultrasound device can drop rapidly below the minimum detectable signal level (MDSL), even for reasonable $L$ values. For a 1-mm-thick steel wall, for example, only 12.47% of its original magnitude passes through the wall at $f = 500$ kHz, resulting in a total energy drop ($T_B^4$) of 36.16 dB. The contours in Fig. 1c provide further insights. Imaging without using the RTP for $L$ less than 20 m is not an issue for fre-quencies ranging from 500 to 1500 kHz when there is no barrier; however, almost all areas in $L$-$f$ plane considered in the figure cannot be accessible if there is a barrier (see Methods). As the frequency range for imaging may be limited in actual application, a method of full ultrasonic wave transmission through a barrier at any (i.e., lower in this case) desired frequency is indispensable for imaging an object behind an impenetrable barrier. We provide barrier-through ultrasonic ima-ging using our RTPs. The underlying physics of our imaging approach is based on the FP resonance phenomenon[26,27]; full transmission is possible through a barrier at certain frequencies depending on the

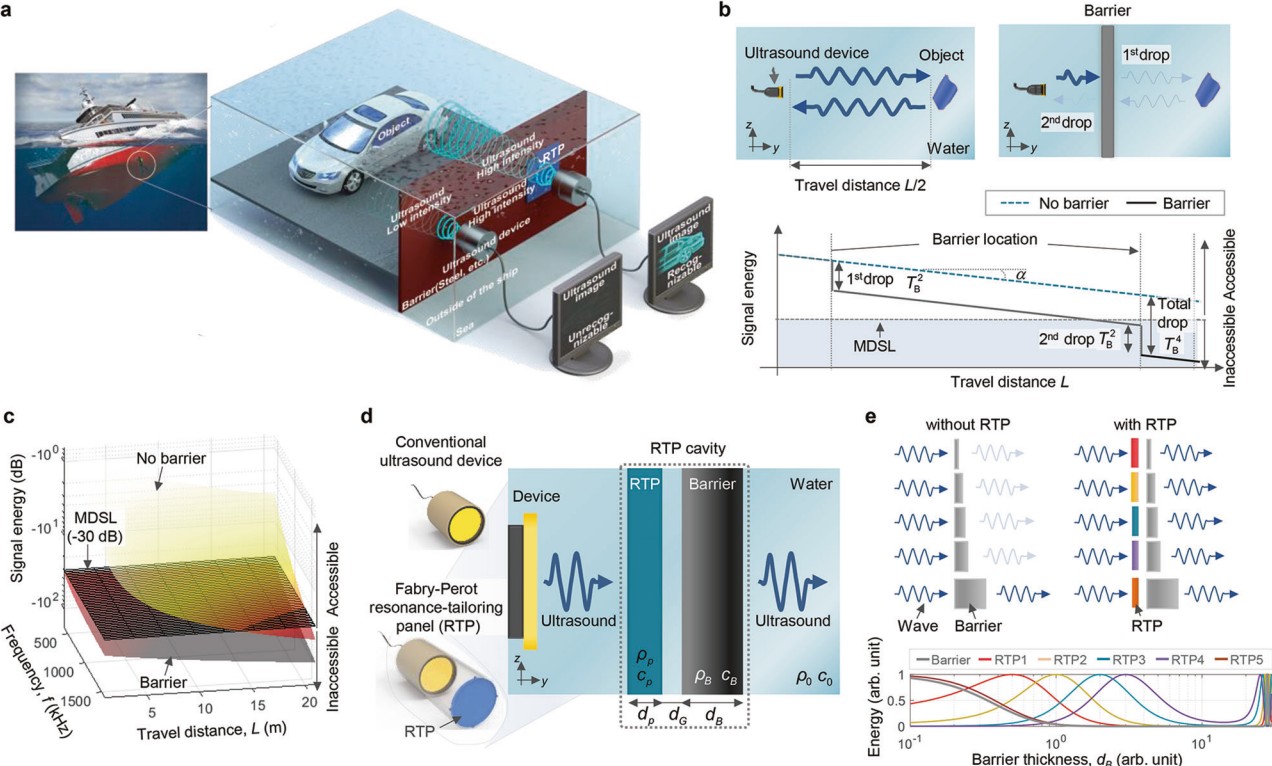

**Fig. 1 | Ultrasonic imaging through a nearly impenetrable barrier and concept of Fabry-Perot resonance-tailoring panel (RTP) for possible full transmission through the barrier. a** Schematic of ultrasonic scanning inside a sunken vessel. **b** With the schematic configurations on top, the signal energy plot regarding the travel distance $L$ is presented. A blue dashed line and a grey solid line respectively denote cases with and without a barrier. The energy decreases by the square of transmittance $T_B$ twice due to the barrier, and consequently, the signal energy falls even below a minimum detectable signal level (MDSL). **c** Accessible range plotted in the frequency-distance domain. The colour bar represents the signal energy. **d** Schematic of the RTP with possible application configuration on the left. The RTP is installed in front of the barrier at a controlled distance $d_G$. **e** The transmitted energy regarding the barrier thickness $d_B$. The solid grey line denotes the without barrier case while, red, yellow, blue, purple, and brown lines represent the cases where properly designed RTPs are installed at each controlled distance $d_G$.

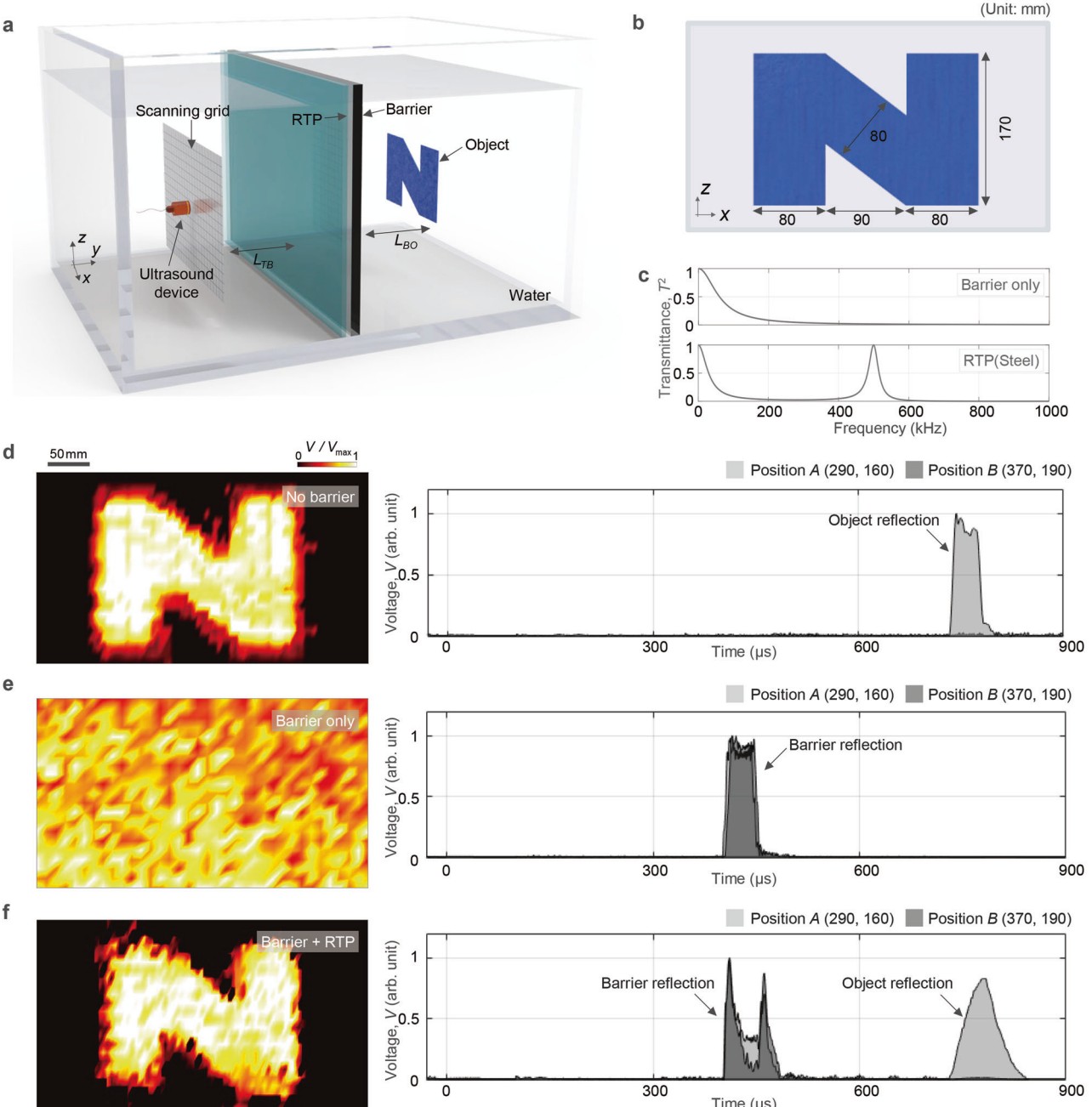

**Fig. 2 | Ultrasonic imaging experiments in water at 500 kHz. a** Schematic of the experimental setup including a 1-mm-thick steel barrier (density $\rho_B = 8000$ kg·m$^{-3}$, Young's modulus $E_B = 200$ GPa, and Poisson ratio $\nu_B = 0.29$). Here, $L_{TB} = 0.30$ m and $L_{BO} = 0.25$ m. **b** A geometry of the object. **c** Transmittance curves of the barrier and RTP. **d**–**f** C-scanned images. The cases of no barrier (top), barrier only (middle), and RTP installed in front of the barrier (bottom) are shown. The voltage amplitude of the time signal is denoted as $V$, while $V_{max}$ represents the maximum value of $V$ along the contour. The plot on the right side of (**d**–**f**) compares the signal envelopes received at two different locations, $P(x, z) = A$ (290, 160) and $B$ (370, 190) (unit: mm), respectively, as a function of time. The object is presented only at Position $A$.

thickness of a barrier ($d_B$ in Fig. 1d). However, its use in imaging may be critically limited because this phenomenon occurs only at discrete frequencies. For instance, a steel barrier of $d_B = 1$ mm fully transmits ultrasound at 2862 kHz, which is the fundamental FP resonance frequency. Note that a wave at 2862 kHz travelling a 10 m ($L = 10$ m) experiences attenuation over 1788 times compared with that of a wave at 500 kHz[24] in water. To ensure that the measured signal magnitude for imaging is larger than the MDSL, the imaging frequency may be reduced to 500 kHz, for example. However, the barrier prevents sufficient transmission of energy at this frequency. Our approach for quality imaging utilises the FP resonance phenomenon at any desired frequency, unlike the intrinsic FP resonance frequencies. Here, we

devise an RTP of a certain material and thickness for imaging through a barrier and position it at a controlled distance in front of the barrier. This allows for nearly full transmission through a system composed of the RTP, the water gap, and the barrier at any desired frequency, including frequencies lower than the fundamental RP resonance frequency.

Figure 1d illustrates the installation of the RTP in front of the barrier. If the RTP is absent, the setup represents a conventional ultrasonic imaging scheme. We may view the RTP and the barrier as partially transparent mirrors entailing phase shifts, which can be regarded as forming an FP cavity[28–30]. Supplementary Fig. 1 shows cascading reflections between the RTP and barrier at a general

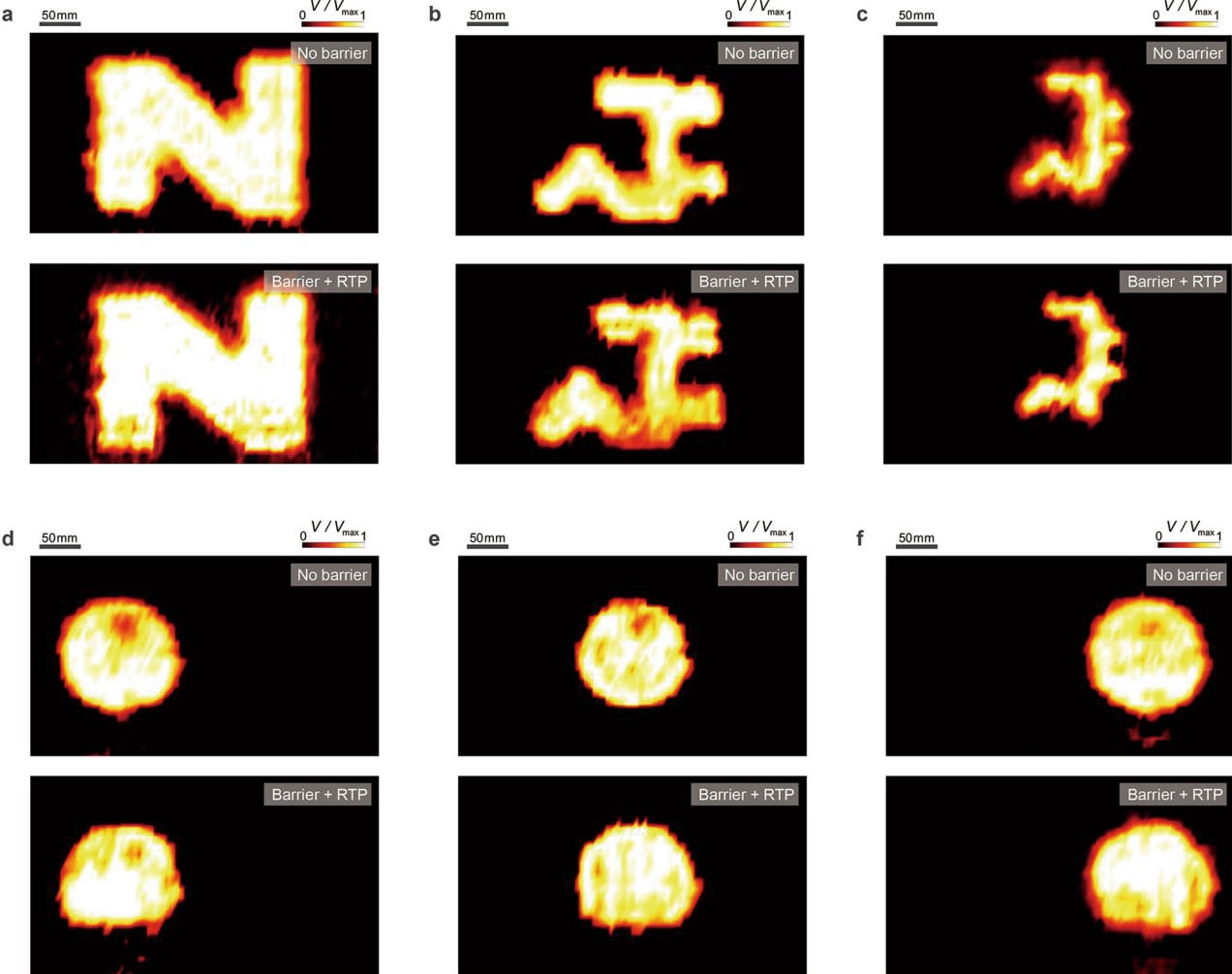

**Fig. 3 | Ultrasonically C-scanned images for complicated shaped objects and circular objects at different locations. a–c** Ultrasound images of complicated shaped objects with different feature widths. **d–f** Target circular objects displaced by 100 mm. Results for the cases of no barrier (top) and RTP installed in front of the barrier (bottom) are plotted. The voltage amplitude of the time signal is denoted as $V$, while $V_{max}$ represents the maximum value of $V$ along the contour.

frequency. Using the wave reflection patterns in the figure and following the derivation in the Methods section, the conditions for full transmission at the desired frequency can be symbolically written as

$$F_M(\omega, \rho_B, c_B, d_B; \rho_p, c_p, d_p) = 0 \tag{1a}$$

and

$$F_P(\omega, \rho_B, c_B, d_B, \rho_p, c_p, d_p; d_G) = 0, \tag{1b}$$

with the angular frequency $\omega = 2\pi f$ ($f$: frequency). One can show (see Methods) that the parameters of the RTP (density $\rho_p$, phase velocity $c_p$, and thickness $d_p$) can be determined first by solving Eq. (1a) for a given barrier ($\rho_B$, $c_B$, and $d_B$). Then, the RTP-barrier gap distance $d_G$ can be sequentially determined from Eq. (1b). One can observe abundant freedom in the design of the RTP because many possible sets of ($\rho_p$, $c_p$, and $d_p$) can satisfy Eq. (1a) for a given barrier. This means that the material of the RTP can be selected from a wide range of materials, and the simplest choice would be to use the same material as that of the barrier. For the experimental results shown in Fig. 2 for a 1-mm-thick steel plate barrier at 500 kHz, the RTP was made of the same material as the barrier, while the gap distance was adjusted to $d_G = 0.051$ mm. Supplementary Fig. 2a presents different design options and their transmittances. Supplementary Table 1 presents

RTPs for a different set of barriers. To address the flexibility of the RTP approach in choosing the operating frequency, Fig. 1e compares the wave transmissions through a barrier with and without the RTP. Without RTP, full transmission occurs only at a discrete FP resonance frequency intrinsic to the thickness of the barrier.

### Ultrasonic imaging through a nearly impenetrable barrier
We demonstrate ultrasonic imaging through a barrier using RTP. We specifically concentrated on recognising the intricate shape of the object. Experiments were performed in an acrylic tank shown in Fig. 2a and Supplementary Fig. 3a. Water is filled in a tank and the object depicted in Fig. 2b is positioned behind a barrier. For the barrier material, 1-mm-thick steel was chosen. The images were scanned at 500 kHz considering the diffraction limit ($\approx \lambda/2 = 1.5$ mm) to sufficiently capture the details of the objects. Note that 500 kHz is much lower than the lowest intrinsic FP resonance frequency of the barrier, 2826 kHz. The transmittance over the barrier is 1.56%. To increase the transmittance, an RTP the based material of which is stainless steel was designed. The frequency-dependent transmittance in Fig. 2c shows that our RTP realises full transmission at 500 kHz. For a fair comparison, the same signal processing technique was used for all the experiments (see Methods).

Figure 2d–f compare the C-scanned images (i) without any barrier, (ii) with the barrier only, and (iii) with the barrier and the RTP. The

nominal C-scanned image obtained without any barrier provides geometric details. The geometries of the used objects are given in Supplementary Fig. 4. On the contrary, the C-scanned image obtained with the barrier in front of the object provides virtually no information on the geometries, demonstrating that imaging an object behind a nearly impenetrable barrier is unfeasible. The C-scanned image, when the RTP is placed in front of the barrier, provides geometric details nearly comparable to those of the nominal C-scanned image. To quantitatively compare the image quality, we use the structural similarity index measure (SSIM)[31] ($0 \leq$ SSIM $\leq 1$, 0: no similarity, 1: full similarity), which estimates the similarity between two images; the SSIM is useful in comparing the perception between two images than other measurement methods[31]. Specifically, the SSIMs for the images shown in Fig. 2d–f with respect to the exact geometries are as follows:

$$\text{SSIM}_{\text{No barrier}} = 0.6148, \text{SSIM}_{\text{Barrier}} = 0.0237, \text{SSIM}_{\text{Barrier + RTP}} = 0.5708.$$

These values indicate that C-scanning using the RTP recovers nearly the same image information as the nominal image information because $\text{SSIM}_{\text{Barrier + RTP}}$ differs from $\text{SSIM}_{\text{No barrier}}$ by only 7.16%. The envelopes of the measured time signals for the target positions $P(x, z) = A$ (290, 160) and $B$ (370, 190) (unit: mm) are shown on the right side. The object is present only at Position $A$. The restored amplitude of object reflection in Fig. 2f enables barrier-through imaging. The three-dimensional contours and B-scan contours that measure the signal along a single trajectory (one-dimensional plots) are presented in Supplementary Fig. 5, from which the two horizontal (at $z = 50$ and 170 mm) and two vertical lines (at $x = 180$ and 270 mm) can be delineated.

Motivated by the imaging results in Fig. 2, ultrasonic imaging for different types of objects was conducted with an automated scanning system in a larger water tank ($135 \times 110 \times 60$ cm$^3$). The experimental setup is depicted in Supplementary Fig. 3c and the geometries of the used objects are given in Supplementary Fig. 4. A thicker barrier plate (4 mm, aluminium) was opted to reduce the adverse effects of plate bending. The same 4-mm-thick aluminium plate was designed as an RTP to enable RTP-harnessed full transmission at 100 kHz and 700 kHz, which are lower than the barrier's FP resonance frequency of 800 kHz (see Supplementary Fig. 2b). The gap between the RTP and the barrier was equalled to 0.937 mm. To preserve the details of the smallest feature scale, 700 kHz was used for experiments. Figure 3a–c show the C-scanned images of various objects. Among other things, the widths of the features were reduced from 80 mm in Fig. 3a to 50 mm in Fig. 3b and 30 mm in Fig. 3c. Figure 3 demonstrates that the scanned images captured through the RTP-harnessed barrier are almost identical to those captured without the barrier, exhibiting intricate geometrical details of thin features. The differences between the nominal and RTP SSIM values are less than 3%, confirming this observation (see Supplementary Table 2).

We further demonstrate the location estimation of the object with circular objects. A 2-mm-thick steel plate was used to fabricate the objects. Each object was horizontally moved 100 mm (along the x-axis). C-scanned results are presented in Fig. 3d–f and the estimated centres are given in Supplementary Table 3 (see Methods for calculation). The calculated centres of the circular objects by the RTP method are displaced from their actual locations by 3.0–6.6 mm, whereas those estimated by direct measurements without any barrier are displaced by 1.5–2.0 mm. The estimation errors regarding the object diameter are between 2% and 4%, which compares favourably to the error of 1% in the absence of a barrier.

## Ultrasonic beam transmission through a barrier using RTPs
Ultrasonic beam transmission through impenetrable barriers is one of the other promising applications of our RTPs including medical[12, 32,33] and NDE[15] fields. We numerically demonstrate the effectiveness of the RTPs in terms of ultrasonic beam transmission through a barrier by simulations. For the ultrasonic beams, an unfocused (linear array) beam and a focused beam (curved array) which are typically used in medical and NDE fields were considered; the beams were supposed to penetrate a 1-mm-thick steel barrier (see Methods). Figure 4 shows the ultrasonic beam simulation results for the two representative beams. Each right and left figure respectively show the simulation without and with our RTP. The existence of a barrier severely hampers the wave energy to be transmitted to the left side. Without RTP, the transmittance was reduced to 1.57% (linear array) and 1.55% (focused array), but with RTP, the transmittance reached 91.01% (linear array) and 40.11% (focused array). Although the transmittance enhancement by the RTP for the focused array is somewhat diminished due to the current RTP design based on the assumption of plane-wave propagation, the increase in transmittance from 1.55% to 40.11% is still significant. We further tested the RTPs for barriers of different base materials. The results are presented in Supplementary Fig. 6 and Supplementary Table 4. Each case witnesses drastic transmission enhancement through the barrier.

## Discussion
In this study, we perform ultrasonic imaging of objects behind nearly impenetrable barriers, such as imaging through sunken vessels or human skulls, which remains unresolved. Our technique for ultrasonic imaging of an object behind a metal barrier can pave the way for

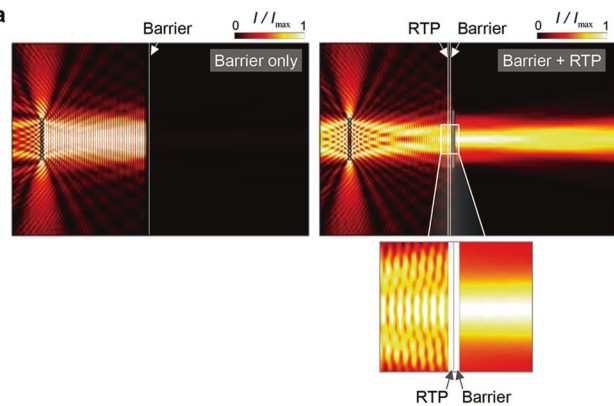
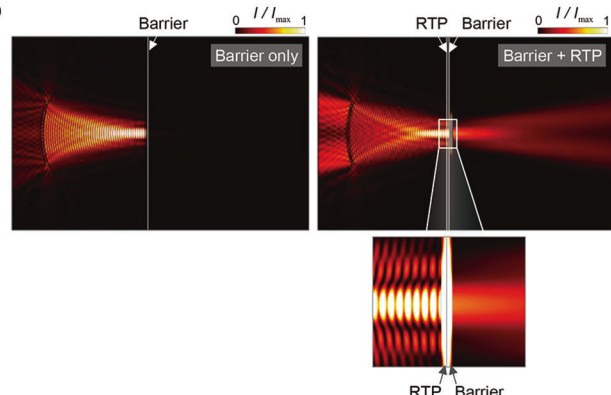

**Fig. 4 | Simulations of ultrasound beam transmission through a 1-mm-thick steel plate barrier with (left) and without (right) RTP at 500 kHz.** Intensity plots of the ultrasound beams: (**a**) Unfocused beam. **b** Focused beam. The RTP is composed of a 1-mm-thick steel plate. The transmitted wave intensity through the barrier is drastically increased by the RTP. The transmitted energy is less increased for the focused beam due to the deviations from the plane wave assumption.

noninvasive ultrasonic imaging of the brain or the inside of a sunken vessel. Our method was experimentally validated by the successful imaging of submerged objects with relatively complex shapes that were concealed behind a 1-mm-thick steel plate barrier and a 4-mm-thick aluminium plate barrier.

At a frequency different from the FP resonance frequency of the barrier, the barrier-through imaging in water was successfully accomplished. When a barrier's FP resonance frequency is high, wave attenuation can make barrier-through imaging at that frequency challenging. Now, these severe restrictions can be overcome if the proposed RTP is utilised to lower the imaging frequency.

However, certain scientific and technical issues must be resolved before this strategy can be utilised in practical applications. First, high-resolution imaging would require a method to tune the target frequency to over 2 MHz, such as an automated fine gap tuning method. For instance, mechanical gap tuning may be possible if a module-type RTP suggested in Supplementary Note 1 can be used. Alternatively, solid films can be considered for the gap tuning, as suggested in Supplementary Note 2. If an oblique incidence should be considered, some modifications of the RTP may be considered, as discussed in Supplementary Note 4. For better temporal localisation, the barrier-through transmission of pulse signals should be studied further, rather than harmonic waves; a general imaging method that does not rely on the plane-wave assumption must also be developed. In addition, a method is required to counteract the material damping, inhomogeneity, and non-uniformity of barriers. Despite these unanswered questions, it is anticipated that the experimental findings of this study will serve as a new paradigm for imaging through nearly impenetrable barriers.

## Methods

### Calculating signal attenuation for ultrasound detection

Under the assumption of the far field, the equation for an acoustic wave from a cylinder-type wave source travelling across a distance $L$ at a frequency $f$ and angular frequency $\omega = 2\pi f$ can be described in terms of the pressure $P$ as[34]

$$P(f, L, t) = P_0 10^{-\eta \alpha L/10} \frac{\exp(i\omega t - ikL)}{4\pi L}, \quad (2)$$

where $P_0$ is the amplitude, $k$ is the wavenumber, and $\eta$ is a coefficient that converts the sound absorption $\alpha$ into an attenuation scale (dB·m⁻¹). Attenuation $\alpha$ in seawater ($T = 4\,°C$, $P = 1\,atm$) can be described as[24]

$$\alpha(f) = A_1 f_1 f^2/(f_1^2 + f^2) + A_2 P_1 f_2 f^2/(f_2^2 + f^2) + A_3 P_2 f^2, \quad (3)$$

where $A_1 = 1.12 \times 10^{-8}$, $A_2 = 5.92 \times 10^{-8}$, $A_3 = 4.7 \times 10^{-14}$, $f_1 = 7.92 \times 10^2$, $f_2 = 7.07 \times 10^4$, $P_1 = 0.9990$, and $P_2 = 0.9996$. These parameters were calculated to adjust the experimentally obtained attenuation in the reference[24]. The relative signal energy (SE) for a wave with a travelling distance of $L$ can be written as

$$SE_{\text{No barrier}}(L, f) = 20\log_{10} \frac{|p(f, L, t)|}{|p(f, L_0, t)|} \approx -2\eta\alpha(f)L \\ -20\log_{10}(L/L_0) \approx -2\eta\alpha(f)L. \quad (4)$$

For the calculations, we used $\eta = 8.686$[24]. The subscript 'No barrier' refers to the case without a barrier. Here, $L_0$ is chosen to be $L_0 = D^2 f/4c_0$ to describe a dimension characterising the near-field. If we use $D = 60\,mm$, $f = 500\,kHz$, and $c_0 = 1500\,ms^{-1}$, $L_0$ will be 0.3 m. The values of $L_0$ will be different depending on the frequency, we calculated it for 500 kHz as the representative value. The final expression in Eq. (4) is based on the assumption of the travelling distance $L$ that $L \gg L_0$ and $L \gg 1$.

We used Eq. (4) to evaluate the signal energy in Fig. 1b and the contour in Fig. 1c. We used the transmission parameter $\beta(f)$ to quantify the wave signal drop caused by a barrier. This can be expressed as

$$\beta(f) = 10\log_{10}(T_B^4), \quad (5a)$$

where the transmission coefficient $T_B$ can be calculated as[34]

$$T_B = (\cos\varphi_B + 0.5i(Z_0 Z_B^{-1} + Z_0^{-1} Z_B)\sin\varphi_B)^{-1}, \quad (5b)$$

in terms of the impedance $Z$, phase shift $\varphi = kd$, and wavenumber $k$. The subscripts $B$ and $0$ refer to the barrier and background medium, respectively. In the present case, the background medium is water. It should be noted that, during imaging, a wave encounters the barrier twice during the round trip to and from the object (see Fig. 1b). Thus, when a barrier exists, the signal energy can be expressed as

$$SE_{\text{Barrier}}(L, f) = SE_{\text{No barrier}}(L, f) + \beta(f) \quad (6)$$

The contour plot in Fig. 1c is based on (6). For the plot, a 1-mm-thick steel wall was considered as the barrier. The contours in Fig. 1c were prepared for the cases with and without a barrier. For the plot, the MDSL was assumed to be −30 dB.

### Design of RTP based on multiple scattering analysis

We assume that the RTP and barrier are placed in the water at a distance of $d_G$. The subscript $G$ denotes the water gap between the RTP and barrier. When an incident wave encounters the RTP, a portion of this wave is reflected (described by the reflection coefficient $R_p$), whereas the remaining portion is transmitted (described by the transmission coefficient $T_p$) through the RTP (see Supplementary Fig. 1). The same phenomenon occurs when a wave is incident onto a barrier. These transmission and reflection coefficients are calculated by using the transfer matrix method and can be expressed as[34]

$$R_m = 0.5i\bar{Z}_{0m}\sin\varphi_m(\cos\varphi_m + 0.5i\tilde{Z}_{0m}\sin\varphi_m)^{-1}, \quad (7a)$$

$$T_m = (\cos\varphi_m + 0.5i\tilde{Z}_{0m}\sin\varphi_m)^{-1}, (m = p, B), \quad (7b)$$

where subscript $0$ represents the background medium (water), and subscripts $p$ and $B$ denote the RTP, barrier, respectively. The symbols $\tilde{Z}_{ij}$ and $\bar{Z}_{ij}$ are defined as $\tilde{Z}_{ij} = Z_i Z_j^{-1} + Z_i^{-1} Z_j$ and $\bar{Z}_{ij} = Z_i Z_j^{-1} - Z_i^{-1} Z_j$, respectively, which can be explicitly written in terms of the impedance $Z$, phase shift $\varphi = kd$, and wavenumber $k$. The wave transmitted through the RTP experiences a phase shift when it further travels across the gap $d_G$, resulting in the transmission coefficient of $T_p e^{-j\varphi_G}$, where $\varphi_G$ is the phase change through the gap.

The partially transmitted wave through the barrier has a transmission coefficient of $T_p T_B e^{-j\varphi_G}$, whereas the reflected wave from the barrier has a reflection coefficient of $T_p R_B e^{-j\varphi_G}$. The reflected wave that travels back to the RTP has a coefficient of $T_p R_B e^{-2j\varphi_G}$. Thereafter, it is split into two waves—the reflected and transmitted waves—due to the RTP, with coefficients of $T_p R_p R_B e^{-2j\varphi_G}$ and $T_p^2 R_B e^{-2j\varphi_G}$, respectively. These scattering cascades occur infinitely. Therefore, the total reflection and transmission coefficients, $R$ and $T$, can be expressed as

$$R = R_p + \sum_{n=1}^{\infty} T_p^2 R_p^{n-1} R_B^n e^{-2nj\varphi_G} = R_p + T_p^2 R_B e^{-2j\varphi_G}(1 - R_p R_B e^{-2j\varphi_G})^{-1}, \quad (8a)$$

$$T = \sum_{n=1}^{\infty} T_p T_B R_p^{n-1} R_B^{n-1} e^{-(2n-1)j\varphi_G} = T_p T_B e^{-j\varphi_G}(1 - R_p R_B e^{-2j\varphi_G})^{-1}, \quad (8b)$$

Because the magnitudes of the scattering coefficients $|R_p|$, $|T_p|$, $|R_B|$, and $|T_B|$ are less than unity, the infinite sums appearing in Eqs. (8a) and (8b) converge. On setting $R = 0$ in Eq. (8a) for the full (100%) transmission through the compound system comprising the RTP, gap, and barrier, the following full transmission condition (FTC) is obtained:

$$e^{2j\varphi_G} = \chi(D_p, Z_r), \tag{9}$$

with

$$\chi(D_p, Z_r) = R_p^{-1} R_B (R_p^2 - T_p^2). \tag{10}$$

where $Z_r$ is the relative impedance of the RTP with respect to the barrier ($Z_r = Z_p/Z_B$), and $D_p$ is the dimensionless thickness of the RTP with respect to the wavelength in the RTP ($D_p = d_p/\lambda_p$). Equation (9) indicates that the left side of the FTC is determined solely by $\varphi_G$, that is, $d_G$. However, the right side of the FTC is solely described by $D_p$ and $Z_r$. Using these observations, we attempt to solve Eq. (9) in terms of its magnitude and phase, as follows:

*Magnitude*: $1 = |\chi|$

$$F_M(\omega, \rho_B, c_B, d_B; \rho_p, c_p, d_p)$$
$$= 1 - \left|\frac{\bar{Z}_{0B} \sin\varphi_B}{\bar{Z}_{0p} \sin\varphi_p}\right| \frac{\bar{Z}_{0p}^2 \sin\varphi_p^2 + 4}{\sqrt{4\cos^2\varphi_p + \bar{Z}_{0B}^2 \sin\varphi_B^2} \sqrt{4\cos^2\varphi_p + \bar{Z}_{0p}^2 \sin\varphi_p^2}} = 0, \tag{11}$$

*Phase*: $\measuredangle(e^{2j\varphi_G}) = \measuredangle[\chi(D_p, Z_r)]$, i.e., $d_G = \frac{c_0}{2\omega} \tan^{-1}\left(\frac{\text{Im}(\chi)}{\text{Re}(\chi)}\right)$,

$$F_P(\omega, \rho_B, c_B, d_B, \rho_p, c_p, d_p; d_G) = \begin{cases} d_G - \kappa \, (\kappa \geq 0) \\ d_G - \kappa - 0.5 c_0 \pi \omega (\kappa < 0), \end{cases} \tag{12}$$

where $\kappa = 0.5 c_0 \omega^{-1} \tan^{-1}((2\tilde{Z}_{0B} \cos\varphi_p \sin\varphi_B + 2\tilde{Z}_{0p} \sin\varphi_p \cos\varphi_B)(\tilde{Z}_{0p}\tilde{Z}_{0B} \sin\varphi_p \sin\varphi_B - 4\cos\varphi_p \cos\varphi_B)^{-1})$. $F_P$ has different forms to assure the sign of the $d_G$ positive. The advantages of using Eqs. (11) and (12) to solve Eq. (9), instead of using its real and imaginary parts, are apparent because the two equations can be solved sequentially. We use Eq. (11) to determine the thickness ($d_p$) and impedance ($Z_p$) of the desired RTP when the dimension and material of a barrier are available. Equation (12) is then sequentially used to determine $d_G$, i.e., the gap distance. It is also noted that different materials can be chosen for the RTP because only the impedance $Z_p$ (and not specifically the density $\rho_p$ or phase velocity $c_p$) is determined using Eq. (11).

## Experimental details for imaging

Supplementary Fig. 3a shows the experimental setup used to obtain ultrasound imaging results in Fig. 2. The experiments were performed in an open-topped acrylic tank ($150 \times 48 \times 50$ cm³). The barrier and the RTP are made of a 1-mm-thick steel plate and have the same dimension of $450 \times 450$ mm². They were mounted on the acrylic tank by a polymer frame at the centre. The distance $d_G$ was controlled by a micrometre (343–250–30, Mitutoyo, Japan). The scanning was conducted using a multi-axis stage. A scanning area of $400 \times 230$ mm² was meshed with $5 \times 10$ mm² pieces. Supplementary Fig. 3b presents this mesh. In this manner, measurements were performed at $81 \times 24$ points to cover the entire scanning area. An immersion-type transmitter (GS100-D25, The Ultran Group, USA) and a needle-type hydrophone (NH4000, Precision Acoustics, United Kingdom) were mounted on the scanning head. A function generator (33250 A, Agilent Technologies Inc., USA) generated and sent reference signals and a power amplifier (AG1017L, T&C Power Conversion, USA) boosted them before they were sent to the ultrasonic transmitter through coaxial cables. The reflected signals from the object were collected by the hydrophone, amplified via a

submersible preamplifier (HP10, Precision Acoustics, United Kingdom) and recorded by an oscilloscope (WaveRunner 104MXi-A, LeCroy, USA). The time-of-flight calculation method was used to evaluate the arrival time of the reflected signal. The scanned images were constructed using the extracted magnitude of the FFT corresponding to the object reflection arrival.

On the other hand, Supplementary Fig. 3c shows the experimental setup used to obtain the ultrasound images in Fig. 3. Experiments were also conducted in an open-topped acrylic tank ($135 \times 110 \times 60$ cm³). In this case, the barrier and the RTP were made of thicker plates, 4-mm-thick aluminium plates (dimension: $800 \times 450$ mm²). The distance $d_G$ was controlled by a micrometre (343–250–30, Mitutoyo, Japan). In this case, an automated programme in the operating computer was used to control the scanning system and record the measured signals via the receiving transducer[35]. The scanning area and the mesh as given in Supplementary Fig. 3b were used. An immersion-type transmitter (GS100-D25, The Ultran Group, USA) and optical hydrophone (Eta100 L Ultra, Xarion Laser Acoustics, Austria) were mounted on the scanning head. A function generator (33250 A, Agilent Technologies Inc., USA) was employed to send the reference signals and a power amplifier (HAS4052, NF Corporation, Japan) boosted the signal. The ultrasonic signals were measured by the optical hydrophone and amplified via the built-in preamplifier before they were sent to the scanning programme for signal processing. The arrival range was predicted based on a time-of-flight calculation. The programme then extracted the maximum FFT magnitude of the reflection arrival and constructed scanned images. Supplementary Note 5 provides details on the RTP experiment and some preliminary experimental results.

## Barriers, RTP, and objects

The barriers were made of a 1-mm-thick steel plate and a 4-mm-thick aluminium plate. The same steel and aluminium plates were used as the base material of the RTP. The theoretically calculated thickness of the RTP in the present experiment were $d_p = 1.0$ mm and $d_p = 4.0$ mm, while the gap distances were $d_G = 0.051$ mm and $d_G = 0.937$ mm, respectively for steel (500 kHz) and aluminium barriers (700 kHz). All the objects were composed of either 1- or 2-mm-thick steel plates. Specifically, the objects shown in Figs. 2b and 3a–c were composed of 1-mm-thick steel plates, whereas the objects shown in Fig. 3d–f were made of 2-mm-thick steel plates. Different thicknesses were used to facilitate fabrication. The exact geometries of the objects are shown in Supplementary Table 3.

## Calculating centre location of object in images

The centre of the object was calculated in each ultrasonic image using a weighted sum. The centre locations are estimated as:

$$x_c = \sum_{i,j} x_i V(x_i, z_j) / \sum_{i,j} V(x_i, z_j), \quad z_c = \sum_{i,j} z_j V(x_i, z_j) / \sum_{i,j} V(x_i, z_j), \tag{13}$$

where $V(x_i, z_j)$ denotes the amplitude of the image measured at the point $(x_i, z_j)$.

## Transmission simulation

Finite element simulations were conducted to investigate the wave transmission through the barrier, with and without the RTP; this was realised using the acoustic module of COMSOL MULTIPHYSICS 5.3[11]. Water (density $\rho_0 = 1000$ kg·m⁻³ and phase velocity $c_0 = 1500$ m·s⁻¹) was used as the background material to fill the entire domain, which was 200 mm along the horizontal direction and 130 mm along the vertical direction. A plane-wave radiation condition was applied to the perimeter of the domain to prevent reflections from the boundaries. We considered cases involving unfocused and focused beams corresponding to the linear and focused transducer arrays, respectively[11,36]. The linear array comprised nine transducers emanating an ultrasonic

wave as a monopole source with a radius of 1 mm. Two adjacent transducers were located 2.6 mm apart along the vertical direction. The maximum amplitude of the beam occurred 90 mm away from the array. By contrast, the focused array comprised 19 transducers focusing on a point located 60 mm to the right of the array centre. Furthermore, the barrier was located 80 mm horizontally from the array centre. The RTP, which was 40 mm in length along the vertical direction, was installed in front of the barrier. The RTP thickness $d_p$ and distance $d_G$ were calculated using Eqs. (11) and (12).

## Data availability
Data are available from the corresponding authors S.H.M and Y.Y.K. upon request.

## Code availability
The codes used for the theoretical calculations of the transmission in this study are available from the corresponding authors S.H.M. and Y.Y.K. upon request.

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

## Acknowledgements
This research was supported by the Global Frontier R&D Programme on Centre for Wave Energy Control based on Metamaterials (CAMM-2014M3A6B3063711) contracted through the Institute of Advanced Machines and Design at Seoul National University funded by the Korea Ministry of Science, ICT & Future Planning. This research was also supported by the National Research Foundation of Korea (NRF-2022R1A2C2008067) funded by the Korea Ministry of Science, ICT & Future Planning. This work was also supported by the Character-isation Platform for Advanced Materials, funded by the Korea Research Institute of Standards and Science (KRISS – 2022 – GP2022-0013).

## Author contributions

Y.Y.K. initiated and led the project. C.I.P. conceived the RTP system. C.I.P. and W.L. analysed and developed the RTP for ultrasound imaging. C.I.P., S.C., W.C., M.K., Y.Y.K. and H.M.S. designed experiments. C.I.P., W.C. and S.C. performed experiments. H.M.S. supervised experiments. C.I.P. and Y.Y.K. wrote the manuscript.

## Competing interests

The authors declare the following competing interests. Four patent applications have been filed by the Seoul National University, Republic of Korea on the transmission enhancement method and barrier-through ultrasonic imaging by the Fabry-Perot resonance tailoring panel: Korea patent application (10-2020-00653, approved), Korea patent application (10-2021-0164875, approved), PCT (PCT/KR2021/006722, approved), and United States patent application (17/637,754, filed), on which C.I.P and Y.Y.K. are listed as inventors. The remaining authors declare no competing interests.
