## [Peer Review File · Nature Communications]

Ultrasonic barrier-through imaging by Fabry-Perot resonance-tailoring panelReviewer #1 (Remarks to the Author):

This is a very good paper, in that it takes a seemingly simple idea, and yet shows that it can give extremely good results. As the authors acknowledge, the idea of creating a Fabry-Perot (FP) cavity using two parallel barriers (as partial reflectors) is not new, and has been used for many decades in lasers, interferometers and other optical designs. In addition, Helmholtz and Fabry-Pérot resonances have long been used for noise/acoustic control purposes. However, I have not seen this particular use of FP resonances, namely the use of a second barrier to deliberately form a resonance to allow transmission of pulse ultrasonic signals in an underwater environment.

The authors have done a good job of quantifying the performance of their structure in imaging terms. I do have a few questions for the authors, simply to allow readers to more fully understand the processes at work:

The authors state that they use a gap of $dG=0.051$ mm (line 117) between the two barriers to create a FP resonance. Their Extended Data Figure 2 also quotes some very thin gaps between the two barriers. I think a little more explanation should be given to the reader for these very small values, as in reality this would be very difficult to achieve over an extended object (such as the one they show as an illustration at the beginning of their paper). This is presumably not just a resonance of the water gap between the two barriers but of the whole structure. Perhaps a little more insight into this would help the reader - e.g. how such a gap could be maintained across a real barrier, and how any angular tilt between the two barriers might impact on any practical implementation in real life. Is there no way of increasing this gap by changing the thickness/material of the added barrier beyond those considered? Are there any other ways in which this could be achieved?

Other than the above, the paper is very well written, and explanations given throughout. The Extended Data is welcome in order to fully understand the processes at work.

Reviewer #2 (Remarks to the Author):

This article presents a design of Fabry-Perot resonance-tailoring panel for selectively enhance information on the shape and position of objects hidden behind a barrier within a small margin of error for any tailored frequency. While the paper contains exciting experimental results of ultrasonic imaging of objects embedded in the water behind the metal barrier, the manuscript could be strengthened by providing more convincing arguments with respect to the following issues:

1. The transmission distance and signal energy in Figure 1B have unit values, but the corresponding scales are not given on the axes. Is the signal energy attenuation strictly linear with the distance? Figure 1C is a quantitative L-f plot of signal energy. In the Methods section, when calculating signal attenuation, the aperture of the radiation surface must also be considered. In addition to considering the sound absorption coefficient, the far field also needs to consider the expansion loss of the wavefront. As shown in Figure 1D, when ultrasound with a certain radiation aperture enters an infinitely large medium layer, there is also expansion loss of the wavefront besides the sound attenuation inside the medium. If authors approximately regard it as a plane wave, please give a comparison between numerical simulations and experiments.

2. In Figure 1E, the author's idea is whether it is possible to achieve full transmission of multiple thicknesses by using RTP with the same thickness and different material properties for different thicknesses of barriers. In my opinion, it is not clear enough. If I change the frequency of the incident wave, can I easily adapt to barriers of different thicknesses without changing the material and thickness of RTP? What are the benefits

of doing so? Can authors provide a method for using one-single tunable material parameters to adapt the barriers with different thicknesses with the broadband response? As mentioned by the authors in the main text, "Our approach for quality imaging utilizes the FP resonance phenomenon at any desired frequency, unlike the intrinsic FP resonance frequencies". It proves that the design of broadband characteristics may be necessary. A more reasonable application scenario should be to cover the barrier surface with an RTP film to achieve the reduction of underwater backscattering sound energy, thereby achieving concealment and camouflage functions.

Figure 2 refers to an ultrasound imaging experiment at 500 kHz, however, in my opinion, the results in Figure 2C are not necessary because there are not any experimental results here. A possible modification would be to make a comparison of the simulated and experimental results.

3. Figure 3, in my view, is just a repetition of the results of Figure 2. The only difference between Figure 2 and Figure 3 is that the optical hydrophone is used to replace the needle-type hydrophone to obtain the C-scan imaging. However, no broadband experimental results are given to prove the scenarios by the authors in the INTRODUCTION part: "As the frequency range for imaging may be limited in actual application, a novel method of full ultrasonic wave transmission through a barrier at any (i.e., lower in this case) desired frequency is indispensable for imaging an object behind an impenetrable barrier."

4. In the simulation results of Figure 4, the authors want to show that both linear and focused arrays can improve the transmitted acoustic energy by RTP, but no experimental proof and application scenarios are given. More important than the simulation results in Figure 4 is how to experimentally apply the designed RTP to adapt the broadband acoustic transmission with irregular surfaces (e.g., the outer surface of the sunken vessel).

Minor comments:

1. Figure 1B compares the relationship between amplitude attenuation and transmission distance with or without barriers, and it is recommended not to put the unit.

2. Figure 1C demonstrates the necessity of the new method for imaging objects behind an impenetrable barrier in a wide frequency range, but why only perform simulation experiments at 500kHz?

3. Can the authors provide a comparison of theoretical and experimental results for Figure 2C?

4. What does V/V_{max} in Figure 2. d-f refer to? What is the formula for calculating the average voltage in the experiment?

5. What is the fundamental difference between the method in this paper and FP resonance? Is there detailed proof?

Response letter for NCOMMS-23-02985-T

Date: Aug. 1, 2023

Ultrasonic barrier-through imaging by Fabry-Perot resonance-tailoring panel

by Chung Il Park^{1,2}, Seungah Choe^{1,2}, Woorim Lee^{1,2}, Wonjae Choi^{3,4}, Miso Kim⁵, Hong Min Seung^{3,4,*}, and Yoon Young Kim^{1,2,*}

The authors are grateful to the reviewers and the editor for their valuable comments and suggestions on the submitted manuscript of NCOMMS-23-02985-T. The original manuscript has been revised to address the comments and suggestions properly for a possible publication of this paper in *Nature Communications*. We would like to respond to the Reviewer's following comments item by item as follows.

Any updated, modified, or newly incorporated information has been emphasized with a yellow highlight in both the manuscript and this accompanying response letter. Equation and figure numbers are modified as a result of the revision.

Response to reviewer #1.

Reviewer #1:

This is a very good paper, in that it takes a seemingly simple idea, and yet shows that it can give extremely good results. As the authors acknowledge, the idea of creating a Fabry-Perot (FP) cavity using two parallel barriers (as partial reflectors) is not new, and has been used for many decades in lasers, interferometers and other optical designs. In addition, Helmholtz and Fabry-Pérot resonances have long been used for noise/acoustic control purposes. However, I have not seen this particular use of FP resonances, namely the use of a second barrier to deliberately form a resonance to allow transmission of pulse ultrasonic signals in an underwater environment. The authors have done a good job of quantifying the performance of their structure in imaging terms. I do have a few questions for the authors, simply to allow readers to more fully understand the processes at work.

Comment #1

The authors state that they use a gap of $d_G=0.051$ mm (line 117) between the two barriers to create a FP resonance. Their Extended Data Figure 2 also quotes some very thin gaps between the two barriers. I think a little more explanation should be given to the reader for these very small values, as in reality this would be very difficult to achieve over an extended object (such as the one they show as an illustration at the beginning of their paper). This is presumably not just a resonance of the water gap between the two barriers but of the whole structure. Perhaps a little more insight into this would help the reader - e.g. how such a gap could be maintained across a real barrier, and how any angular tilt between the two barriers might impact on any practical implementation in real life. Is there no way of increasing this gap by changing the thickness/material of the added barrier beyond those considered? Are there any other ways in which this could be achieved? Other than the above, the paper is very well written, and explanations are given throughout. The Extended Data is welcome in order to fully understand the processes at work.

Response

[on gap distance]

Maintaining the gap distance over the entire area is important because the failure of the correct gap adjustment could degrade the transmission efficiency. In the actual application of this approach, we may be able to alleviate the issue using a compact module-type RPT, which allows fine-tuning by a linear motor or a screw. The suggested configuration is shown below.

Figure A| Schematic drawing of the RPT module (see *Supplementary Figure 2*.)

Our preliminary tests suggest that the minimum dimension of the module-type RPT should be in the order of $4-5\lambda$, resulting in $15 \times 15 \text{ mm}^2$ (horizontal and vertical lengths of the panel cross section) if the frequency of interest is 500 kHz. We hope to report some experimental results based on the module-type RPT being developed in the near future.

Another approach to facilitate the gap distance control is to replace the water-filled gap with solid films having nearly the same impedance as the water impedance, as shown in Fig. B.

Figure B| RPTs employing different gap materials

Specifically, we can use such as commercially-available ethyl vinyl acetate films (28% Acetate) whose impedance is 1.064 times that of the water. The design principle of these films is exactly the same as that given in the main text. This approach is newly added as *Supplementary Note 2*.

Alternatively, it may also be possible to adjust the material for the RTP if the gap distance (d_G) cannot be altered. One can the desired material properties of the RTP and may realize them using a metamaterial if d_G is fixed. The related explanations are added as *Supplementary Note 3*.

[on the effect of incidence angle]

The explanations of the effects of the incidence angle are given in *Supplementary Note 4*. In the note, we have demonstrated that the transmission efficiency drops if the incidence deviates from the normal incidence. However, modifying the gap distance according to the wave vector in the propagating direction could improve the transmission efficiency very close to the ideal case. The analysis results are presented in Supplementary Note 4; we give the essential results in Figure C below.

Figure C | RPTs for oblique incidence

We can summarize the amendment made to reflect comment #1 as follows.

1. **Supplementary Note 1, pages 2-3 (newly added)**
2. **Supplementary Note 2, pages 4-5 (newly added)**
3. **Supplementary Note 3, pages 6-8 (newly added)**
4. **Supplementary Note 4, pages 9-12 (newly added)**
5. **In the main text (highlighted in yellow)**

Page 9, 2nd paragraph, (newly added):

... such as an automated fine gap tuning method. *For instance, mechanical gap tuning may be possible if a module-type RTP suggested in Supplementary Note 1 can be used. Alternatively, solid films can be considered for the gap tuning, as suggested in Supplementary Note 2. If an oblique incidence should be considered, some modifications of the RTP may be considered, as discussed in Supplementary Note 4. For better temporal...*

Response to reviewer #2.

Reviewer #2:

This article presents a design of Fabry-Perot resonance-tailoring panel for selectively enhance

information on the shape and position of objects hidden behind a barrier within a small margin of error for any tailored frequency. While the paper contains exciting experimental results of ultrasonic imaging of objects embedded in the water behind the metal barrier, the manuscript could be strengthened by providing more convincing arguments with respect to the following issues:

Comment #1

The transmission distance and signal energy in Figure 1B have unit values, but the corresponding scales are not given on the axes. Is the signal energy attenuation strictly linear with the distance? Figure 1C is a quantitative L-f plot of signal energy. In the Methods section, when calculating signal attenuation, the aperture of the radiation surface must also be considered. In addition to considering the sound absorption coefficient, the far field also needs to consider the expansion loss of the wavefront. As shown in Figure 1D, when ultrasound with a certain radiation aperture enters an infinitely large medium layer, there is also expansion loss of the wavefront besides the sound attenuation inside the medium. If authors approximately regard it as a plane wave, please give a comparison between numerical simulations and experiments.

Response

Thank you for the comment. The main motivation to include these figures is to mention the importance of using any frequency for acoustic wave penetration through an opaque barrier, not to explain the detailed wave physics in water. Therefore, a more detailed analysis of the acoustic field (such as the comparison between numerical simulations and experiments) was not performed. Our main objective is to show that an opaque barrier can be acoustically penetrated if an elaborately-tune RTP is employed and to present experimental evidence. Therefore, we focus on how to get an acoustic image of an object behind an opaque barrier.

However, some misleading statements are corrected as follows.

In the revised version:

Page 18, Equations (2) and (4) (modified):

$$P(f, L, t) = P_0 10^{-\eta\alpha L/10} \frac{\exp(i\omega t - ikL)}{4\pi L}, \quad (1)$$

$$SE_{\text{No barrier}}(L, f) = 20 \log_{10} \frac{|p(f, L, t)|}{|p(f, L_0, t)|} \approx -2\eta\alpha(f)L - 20 \log_{10}(L/L_0) \approx -2\eta\alpha(f)L. \quad (2)$$

Page 18, 2nd paragraph (newly added):

...Here, L_0 is chosen to be $L_0 = D^2 f / 4c_0$ to describe a dimension characterizing the near-field. If we use $D = 60 \text{ mm}$, $f = 500 \text{ kHz}$, and $c_0 = 1500 \text{ m/s}$, L_0 will be 0.3 m . The value of L_0 will be different depending on the frequency, we calculated it for 500 kHz as the representative value. The final expression in Eq. (4) is based on the assumption of the traveling distance L that $L \gg L_0$ and $L \gg 1$. We used...

We modified Equation (4) but we still use Figure 1B as it is useful to convey our motivation. The related modifications were made in the Method section.

We have also provided the experimental details, including the measured transducer field, in Supplementary Note 5. The validated frequency response is also given.

Supplementary Note 5, pages 13-16 (newly added):
Experimental details and data for ultrasonic imaging

Page 23, 1st paragraph (newly added):

*... The program then extracted the maximum FFT magnitude of the reflection arrival and constructed scanned images. **Supplementary Note 5 provides the details on the RTP experiment and some preliminary experimental results...***

Comment #2

In Figure 1E, the author's idea is whether it is possible to achieve full transmission of multiple thicknesses by using RTP with the same thickness and different material properties for different thicknesses of barriers. In my opinion, it is not clear enough. If I change the frequency of the incident wave, can I easily adapt to barriers of different thicknesses without changing the material and thickness of RTP? What are the benefits of doing so? Can authors provide a method for using one-single tunable material parameters to adapt the barriers with different thicknesses with the broadband response? As mentioned by the authors in the main text, "Our approach for quality imaging utilizes the FP resonance phenomenon at any desired frequency, unlike the intrinsic FP resonance frequencies". It proves that the design of broadband characteristics may be necessary. A more reasonable application scenario should be to cover the barrier surface with an RTP film to achieve the reduction of underwater backscattering sound energy, thereby achieving concealment and camouflage functions. Figure 2 refers to an ultrasound imaging experiment at 500 kHz, however, in my opinion, the results in Figure 2C are not necessary because there are not any experimental results here. A possible modification would be to make a comparison of the simulated and experimental results.

Response

We understand the concerns of Reviewer #2 and would like to explain the significance of our work and how we could resolve the concerns of the Reviewer.

1. As the RTP works on the Fabry-Perot resonance, it is inevitably frequency-dependent. However, as far as we look through all available papers and patents, there is no report to get an acoustic image through a metal barrier, as presented here. Accordingly, I hope that the Reviewer values our work from this perspective.

2. We concur about the difficulty of fine-tuning the gap distance. Despite this difficulty, we believe there may be some practical approaches to overcome it.

One idea to alleviate the issue is a compact module-type RPT, which allows a linear motor or crew to fine-tuning. The suggested configuration is shown below.

Figure A| Schematic drawing of the RPT module (see *Supplementary Figure 2*.)

Our preliminary tests suggest that the minimum dimension of the module-type RPT should be in the order of $4-5\lambda$, resulting in $15 \times 15 \text{ mm}^2$ (horizontal and vertical lengths of the panel cross section) if the frequency of interest is 500 kHz. We hope to report some experimental results based on the module-type RPT being developed in the near future.

Another approach to facilitate the gap distance control is to replace the water-filled gap with solid films having nearly the same impedance as the water impedance, as shown in Fig. B.

Figure B| RPTs employing different gap materials

Specifically, we can use such as commercially-available ethyl vinyl acetate films (28% Acetate) whose impedance is 1.064 times that of the water. The design principle of these films is the same as that given in the main text. This approach is newly added as *Supplementary Note 2*.

Alternatively, it may also be possible to adjust the material for the RTP if the gap distance (d_G) cannot be altered. One can find the desired material properties of the RTP and may realize them using a metamaterial if d_G is fixed. The related explanations are added as *Supplementary Note 3*.

3. About the broadband issue: as we mentioned above, the RPT is based on the FPR, which is supposed to work perfectly only for the resonance frequency. However, the tradeoff between transmission efficiency and working frequency band can be made, as done in many other applications using FPR. If that is critically necessary, one can find an optimal value of the gap distance and the thickness of the RTP for best performance using an optimization technique. However, we did not pursue this because we aim to focus on the underlying principle of the RPT and the experimental demonstration of our new approach.

4. About Fig. 2C, we think that it is important to convey the technical issue using this figure. With this figure, the overall explanation becomes easier. Therefore, I hope that Reviewer 2 respects our intention.

Comment #3

Figure 3, in my view, is just a repetition of the results of Figure 2. The only difference between Figure 2 and Figure 3 is that the optical hydrophone is used to replace the needle-type hydrophone to obtain the C-scan imaging. However, no broadband experimental results are given to prove the scenarios by the authors in the INTRODUCTION part: “As the frequency range for imaging may be limited in actual application, a novel method of full ultrasonic wave transmission through a barrier at any (i.e., lower in this case) desired frequency is indispensable for imaging an object behind an impenetrable barrier.”

Response

In the Introduction, we did not argue that our method is broadband; we argued that the selected frequency used for the acoustic penetration of an opaque barrier should be arbitrary (in theory), and our RTP method is a new method for the purpose. We also give our responses to the broadband issue in our response to Comment #1.

Besides, we think that Figure 3 is equally important as Figure 2 because we need to show that our methods can be applied to get images of more complex objects, and also that method is capable of providing meaningful information on spatial localization. Furthermore, the frequencies used to obtain the results in Fig. 2 and Fig. 3 are different.

Comment #4

In the simulation results of Figure 4, the authors want to show that both linear and focused arrays can improve the transmitted acoustic energy by RTP, but no experimental proof and application scenarios are given. More important than the simulation results in Figure 4 is how to experimentally apply the designed RTP to adapt the broadband acoustic transmission with irregular surfaces (e.g., the outer surface of the sunken vessel).

Response

1. About the broadband issue, we responded in our responses to Comment #1 and Comment #3.

2. To provide the experimental results supporting the numerical simulation results in Fig. 4 would be best. However, we were not able to afford to perform the experiments at this point. More importantly, the simulation in Fig. 4 is an illustration of further applications of the RTP, while the main results of the RTP are provided in Figure 2 and Figure 3. We hope that we can report more practical results using the RTP in the near future, but in this paper, we would like to focus on the main message that we can now get an image of an object behind an opaque barrier at any desired frequency if elaborately-designed RTP's are used. We hope that Reviewer

#2 evaluates our work based on this new contribution. One more remark: if our method should be applied to get an image of an object behind a curved opaque barrier, we may consider a piecewise version of our RTP, as sketched in Fig. E. However, a more thorough analysis should be carried out to validate this idea.

Figure E| Current development: piecewise RTPs for curved surfaces of varying thickness

Minor comments:

1. Figure 1B compares the relationship between amplitude attenuation and transmission distance with or without barriers, and it is recommended not to put the unit.

2. Figure 1C demonstrates the necessity of the new method for imaging objects behind an impenetrable barrier in a wide frequency range, but why only perform simulation experiments at 500kHz?

3. Can the authors provide a comparison of theoretical and experimental results for Figure 2C?

4. What does V/V_{max} in Figure 2. d-f refer to? What is the formula for calculating the average voltage in the experiment?

5. What is the fundamental difference between the method in this paper and FP resonance? Is there detailed proof?

Response

1. We removed the units from Figure 1B.

2. We performed experiments at two frequencies: 500 kHz (Figure 2) and 700 kHz (Figure 3).

3. Sorry, the suggested work is out of scope. Please note that our paper is focused on barrier-through imaging.

4. We apply normalization (V/V_{max}) to each imaging and time dataset. See Extended Data Fig. 5 for details.

5. If the RTP is used, our method can be applied for any frequency. However, the direct use of the Fabry-Perot resonance allows the use of certain pre-determined frequencies depending on

the thickness of the barrier. The detailed accounts were given in the Introduction and using a few figures in the main text.

Reviewer #1 (Remarks to the Author):

The paper is now improved and can be published.